# Prevalence of CVD Among Indian Adult Population: Systematic Review and Meta-Analysis

**DOI:** 10.3390/ijerph22040539

**Published:** 2025-04-01

**Authors:** Mohd Shannawaz, Isha Rathi, Nikita Shah, Shazina Saeed, Amrish Chandra, Harpreet Singh

**Affiliations:** 1Amity Institute of Public Health & Hospital Administration, Amity University, Noida 201301, India; isha.rathiph@gmail.com (I.R.); nikita.shah.j@gmail.com (N.S.); ssaeed@amity.edu (S.S.); 2School of Pharmacy, Sharda University, Greater Noida 201310, India; 3Development Research, Indian Council of Medical Research (CMR), Delhi 110029, India; hsingh@bmi.icmr.org.in

**Keywords:** cardiovascular disease, systematic review, meta-analysis, prevalence, India

## Abstract

Cardiovascular disease is among the leading causes of mortality and morbidity globally. Over three-quarters of CVD-related deaths now occur in low- and middle-income countries (LMICs); India accounts for one-fifth of global CVD-related deaths, especially among the younger population. The objective of this study was to evaluate the prevalence of CVD among the Indian adult population. We systematically searched Scopus and PubMed from January 2000 to December 2024 to identify relevant articles and pooled the prevalence of CVD using random-effects meta-analysis. Among the 14,647 records screened, 501 full-text articles were assessed for eligibility and 15 studies were included in the final analysis. The pooled prevalence of CVD was 11% (95% CI: 0.09–0.17). Subgroup analysis showed prevalence rates of 12% among males and 14% among females. Urban areas had a higher prevalence (12%) compared to rural areas (6%), with a significant difference. Our study shows the significant prevalence of cardiovascular disease (CVD) in India, particularly in urban areas, with slightly higher rates among females. Focused public health strategies are required to mitigate the growing burden of CVD, along with preventive measures, to reduce further increases in disease prevalence and related fatalities.

## 1. Introduction

Cardiovascular diseases (CVDs) account for the highest number of deaths worldwide [1]. They were responsible for 20.5 million deaths in 2021, comprising about one-third of all global deaths, a sharp rise from the 12.1 million deaths recorded in 1990 [2,3].

Estimates indicate that non-communicable diseases (NCDs), such as cardiovascular diseases (CVDs), various cancers, chronic respiratory conditions, and diabetes, account for approximately 60% of all deaths [4]. NCDs have become the leading cause of death in India, with cardiovascular diseases being the most prevalent among them [5]. According to WHO (World Health Organisation), in India, NCDs accounted for 63% of all fatalities in 2016, with CVDs accounting for 27% of those deaths [6]. Although cardiovascular disease (CVD) is recognised as a major public health issue in India, access to cardiovascular care remains constrained, as reflected by the low rates of detection, treatment, and adherence to evidence-based treatment guidelines within the population [7].

In this present decade, 20% of the world’s population is estimated to become 65 or older, and, by 2030, the number of deaths from CVDs is anticipated to rise from 16.7 million in 2002 to 23.3 million, including a significant increase in low- and middle-income countries like India [8,9]. Older adults in India are more prone to health risks due to factors like socio-economic and cultural characteristics, insufficient nutritional intake, and poor quality of health facilities, along with limited access to healthcare, which results in one of the highest pocket expenditures [10,11,12]. The World Economic Forum and Harvard School of Public Health conducted a study in which India was expected to face economic losses of approximately USD 2.17 trillion from cardiovascular diseases (CVDs) between 2012 and 2030 [13].

The prevalence of cardiovascular disease in India doubled between 1990 and 2016 [14]. Evidence suggests that the risk of cardiovascular disease (CVD) is generally higher in urban areas than in rural areas, highlighting the potential impact of urban lifestyles on CVD prevalence. Urbanisation is associated with lifestyle modifications, including changes in dietary habits and decreased physical activity, which contribute to increased rates of obesity, hypertension, and diabetes, which are major risk factors for CVD [15].

Cardiovascular disease (CVD) is a significant public health concern in India. Moreover, there is a dearth of research that estimates the overall prevalence of CVD among adults in India through systematic reviews and meta-analysis. Due to a lack of comprehensive data on CVD prevalence in India, health professionals and policymakers are unable to fully understand the magnitude of the problem. Detailed information on the prevalence of cardiovascular diseases (CVD) is crucial for planning and implementing effective preventive strategies. To address this knowledge gap, this study was conducted using data from the scientific literature to provide a comprehensive insight into the prevalence of CVDs among the adult population of India.

## 2. Methodology

### 2.1. Registration

This review was registered with the Prospective Register of Systematic Reviews (PROSPERO: CRD42024627165) and adhered to the PRISMA 2020 guidelines (Preferred Reporting Items for Systematic Literature Reviews and Meta-Analysis) (Appendix A).

### 2.2. Data Sources and Search Strategy

We systematically searched PubMed and Scopus to identify studies reporting the prevalence of CVD among the adult population in India from January 2000 to December 2024. The search strategy is detailed in Appendix A. In our study, we followed the World Health Organisation (WHO) definition of cardiovascular disease (CVD), which encompasses the following: coronary heart disease (CHD), peripheral arterial disease, cerebrovascular disease, rheumatic heart disease, deep vein thrombosis, congenital heart disease, and pulmonary embolism [1].

### 2.3. Inclusion and Exclusion Criteria

A defined set of inclusion and exclusion criteria was applied to identify and select relevant articles for this review.

The criteria for inclusion were as follows: (1) studies related to the prevalence of CVD among the adult population in India; (2) articles written in English; (3) articles published from January 2000 to December 2024; (4) full-text articles available; (5) observational studies, including cross-sectional, cohort, and prospective studies.

The exclusion criteria included the following: (1) studies involving non-human subjects, research focused on children, and other biomedical studies that reported the prevalence of CVD; (2) prevalence reported on individuals with pre-existing systemic conditions such as diabetes or hypertension; (3) articles published in languages other than English and those published prior to January 2000; (4) non-primary articles, including letters to the editor, book chapters, and case reports.

### 2.4. Study Selection

One author (I.R.) exported the included articles from the databases and then imported the articles to Rayyan to identify duplicates and for screening. The articles that were imported were independently screened by two authors (I.R. and N.S.), with any conflicts addressed by a third researcher (M.S.). The full texts of potentially eligible studies were retrieved and assessed for inclusion in the systematic review by two authors.

### 2.5. Data Extraction

The following information and data of the included studies were extracted: author, publication year, gender, age group of the participants, sample size, study area (urban/rural), study design, sampling method, main outcome, criteria for diagnosis of CVD, and the prevalence of CVD. The data were independently extracted by two reviewers using a standardised form.

### 2.6. Quality Assessment of Included Studies

The quality of the included studies was evaluated using the Joanna Briggs Institute (JBI) Critical Appraisal Checklist for Studies Reporting Prevalence Data. The JBI checklist contained 10 questions for each study [16,17]. The answers to questions were “yes”, “no”, or “unclear” and each question scored 1 point for yes and 0 points for no, unclear, or not applicable, leading to a total score ranging from 0 to 10. The scores were then converted to percentages and categorised into three levels of risk of bias: a high risk of bias was indicated if 20–50% of the questions were answered “yes”, a moderate risk of bias if 50–80% were answered “yes”, and a low risk of bias if 80–100% were answered “yes”.

### 2.7. Statistical Analysis

The overall pooled prevalence along with the corresponding 95% confidence intervals (CIs) for the included studies were calculated using a random-effects meta-analysis. The random-effects model assumed that the studies included were random samples drawn from a larger population, allowing the results to be generalised beyond the selected studies [18]. Studies were classified by the gender and geographic location of the participants. Statistical heterogeneity was assessed using I^2^ statistics and Cochran’s Q test and further analysed through meta-regression and stratified analyses based on the gender and geographic location of the study population. A visual inspection of the generated funnel plot was performed to assess its symmetry and determine the potential influence of publication bias on the findings. The meta-analysis was performed using Stata 14.0.

## 3. Results

### 3.1. Study Selection

Figure 1 illustrates the study selection process. We retrieved 14,647 records and, after removing the duplicates and irrelevant articles, examined the titles and abstracts of the remaining 5457 papers. Finally, after reviewing the full text of the remaining 501 papers, we identified 15 studies suitable for meta-analysis.

### 3.2. Study Characteristics

The key characteristics of the included studies are summarised in Table 1. The studies collectively included a total sample size ranging from 500 to 385,055 participants, representing both genders. Most studies included participants aged 20 years and older, while certain longitudinal studies specifically focused on populations aged 45 years and above. Most of the studies were conducted with both rural and urban populations (twelve out of fifteen), while three studies were conducted with rural populations. Samples were selected using multistage cluster sampling in nine studies; other studies employed methods such as simple random sampling, systematic random sampling, and convenience sampling, while one study did not report its sampling method. The studies reported various types of cardiovascular diseases (CVD)s: four studies focused on stroke, four on coronary heart disease (CHD), one on peripheral artery disease (PAD), and one on coronary artery disease. The remaining studies reported combined CVD outcomes. The prevalence of CVD reported in the studies ranged from 0.26% to 35.2%. The studies reported overall prevalence, while some also provided data by gender, urban/rural variations, and type-specific CVD prevalence.

### 3.3. Study Quality Assessment

The quality of the studies was assessed by two authors (N.S. and I.R.) using the Joanna Briggs Institute (JBI) critical appraisal checklist for studies reporting prevalence data. Any disagreements were resolved through consultation with a third reviewer (M.S.). The assessment results, as indicated in Table 2, show, the methodological quality and risk of bias for the individual studies, categorised as low (80–100% scores, marked with green) or moderate (50–80% scores, marked with yellow) risks of bias. Of the 15 articles included in this review, 14 were of high quality with a low risk of bias [19,20,21,22,23,24,25,29,30,31,32,33], while one study [26] had a moderate risk of bias.

### 3.4. Pooled Prevalence of Cardiovascular Diseases (CVDs)

The overall pooled prevalence of CVDs was 11% (95% CI: 0.090–0.120), with severe heterogeneity (I^2^ = 99.97%, Cochran’s Q-statistic *p* < 0.001). This finding suggests that, on average, 11% of the adults in India had CVDs.

The forest plot illustrated in Figure 2 presents the prevalence of cardiovascular disease (CVD) based on a random-effects model.

### 3.5. Subgroup Analysis by Gender of Study Participants

A subgroup analysis based on participants’ gender was conducted to examine potential differences (Figure 3). The weighted pooled prevalence among males was 12% (95% CI: 0.060–0.180), with significant heterogeneity observed across studies (I^2^ = 99.88%, Cochran’s Q-statistic *p* < 0.001). Similarly, the pooled prevalence among females was slightly higher at 14% (95% CI: 0.070–0.210), with severe heterogeneity noted (I^2^ = 99.93%, Cochran’s Q-statistic *p* < 0.001). Although there were numerical differences in prevalence between genders, the test for heterogeneity between subgroups revealed no statistically significant difference (*p* = 0.703). The severe heterogeneity within both subgroups underscored the potential impact of study-specific factors, including population characteristics, study design, and diagnostic criteria, on the prevalence estimates.

### 3.6. Subgroup Analysis by Geographical Location of Study Participants

A subgroup analysis was conducted to assess the prevalence of the condition among urban and rural participants separately (Figure 4). The prevalence of CVD and its associated risk factors varies significantly between urban and rural areas in India, primarily due to differences in lifestyles, as evident in this study [34]. The pooled prevalence among urban participants was 12% (95% CI: 0.090–0.140), with notable heterogeneity observed across studies (I^2^ = 99.92%, Cochran’s Q-statistic *p* < 0.001). In contrast, the pooled prevalence among rural participants was 6% (95% CI: 0.050–0.080), also showing significant heterogeneity (I^2^ = 99.93%, Cochran’s Q-statistic *p* < 0.001).

Moreover, the heterogeneity test between the subgroups revealed a statistically significant difference (*p* = 0.00), indicating that the observed prevalence differences between urban and rural populations may represent meaningful variability.

### 3.7. CVD Prevalence: Assessment of Temporal Variation

We conducted a meta-regression analysis on CVD prevalence over the study year to evaluate temporal trends. Overall, CVD prevalence increased by 0.33% (*p* = 0.443) per year, indicating a slight upward trend over time (Figure 5). However, this increase was not statistically significant. Additionally, the high heterogeneity observed across studies (I^2^ = 99.98%, Tau^2^ = 0.01342) and the low proportion of variance explained by the study year (adjusted R^2^ = −2.28%) indicated that factors beyond the study year may have influenced the reported prevalence trends.

### 3.8. Assessment of Publication Bias

The funnel plot showed asymmetry, with a higher concentration of smaller studies on one side, suggesting potential publication bias (Figure 6). Since Egger’s test was not performed, we used regression analysis to examine the relationship between effect size (prevalence) and its standard error (SE) to assess potential publication bias. However, the regression results did not reveal a statistically significant relationship (F = 1.21, *p* = 0.2878), and the R-squared value of 0.0703 indicated that the SE explained only 7.03% of the variation in prevalence. Given the absence of strong statistical evidence and weak regression results, no adjustments for publication bias were made, such as the trim and fill method. Therefore, we remained cautious in interpreting the adjusted estimates and refrained from assuming that the adjusted pooled prevalence accurately reflected the true prevalence.

## 4. Discussion

The objective of this study was to determine the prevalence of cardiovascular diseases (CVDs) among adults in India. The prevalence of CVD varied between studies. The findings indicate that the pooled prevalence of cardiovascular diseases (CVDs) was 11% (95% CI: 0.090–0.120) in India; high heterogeneity was observed across the prevalence estimates, which could be attributed to differences in study design, population characteristics, and diagnostic criteria. As a result, the findings were interpreted with caution. Comparing the findings of this study with prior research highlights the significant global burden of CVD, both overall and by specific types, in the general population [35,36,37]. Despite variability across studies, the 11% prevalence rate observed in this study underscores the substantial burden of CVD in India. These findings are consistent with the literature on the increasing patterns of CVD risk factors in the country [4,38]. On the other hand, a study conducted in Canada found that 10.7% of South Asians, 5.4% of Europeans, and 2.4% of Chinese had some form of cardiovascular disease (CVD) [39]. These comparisons highlight the regional and ethnic disparities in CVD prevalence, emphasising the need for tailored public health strategies to address these variations.

A subgroup analysis by gender showed a pooled prevalence of 12% (95% CI: 0.060–0.180) in males and 14% (95% CI: 0.070–0.210) in females. Although there was a significant difference in prevalence between the genders, no significant gender-based difference was found using the heterogeneity test (*p* = 0.703), which suggests that the same burden of CVD was present for both genders. In a different study on the geriatric Indian population, the estimated pooled prevalence of CVDs was higher (38.0% males and 40.9% females), with age and gender differences gradually balancing older cases of CVDs [40]. Globally, the overall prevalence of CVDs is higher in females than in males [41]. This may be attributed to the menopausal transition, which is associated with an increased risk of heart disease [42]. However, several studies [43,44] have highlighted a higher risk of heart disease in males. Recent findings, however, indicate a rising prevalence of heart disease among middle-aged women, alongside a declining trend in males within the same age group [45].

This study revealed a significant difference in the prevalence of cardiovascular diseases (CVDs) between urban (12%) and rural areas (6%), emphasising the impact of lifestyle factors. Urbanisation is strongly associated with lifestyle changes that increase health risks, such as decreased physical activity, poor dietary habits, higher consumption of high-calorie foods, and rising obesity rates. These factors are major contributors to the rising prevalence of CVD, as documented in studies globally [46,47,48,49]. Specifically, urban areas tend to exhibit a higher average body mass index (BMI), greater prevalence of hypertension, and lower physical activity levels [50,51,52]. Additionally, environmental factors in urban regions significantly influence both systolic and diastolic blood pressure, further increasing the risk of hypertension [53].

As the global burden of CVDs continues to rise, hypertension is emerging as a leading contributor, particularly in middle-income countries, where one-third of CVD-related deaths are attributed to hypertension [54]. In India, the shift from infectious to non-communicable diseases has occurred within a relatively short period [38]. Consequently, CVD prevalence is rising, driven by several factors: the epidemiological transition, rapid urbanisation, and the increasing adoption of Western lifestyles. Together, these factors are significantly contributing to the growing burden of CVD in the population [49,55]. Our findings are consistent with previous studies that have consistently found a higher prevalence of cardiovascular diseases (CVDs) in urban areas than in rural regions. These trends emphasise the critical need for public health interventions addressing the effects of urbanisation on lifestyle and cardiovascular health, especially in rapidly urbanising regions.

## 5. Strength and Limitations

This systematic review and meta-analysis provide a clear understanding of the growing burden of cardiovascular diseases (CVDs) in India, with a notably higher prevalence in urban areas. Given the limited number of studies conducted in India, our study is unique in its scope and offers valuable insights into CVD trends. The inclusion of subgroup and sensitivity analyses provides valuable insights into variations in CVD prevalence across different demographic groups and evaluates the impact of heterogeneity on the overall findings. A key strength of this review is the rigorous quality assessment and data extraction processes, independently conducted by two reviewers, which minimised the risk of bias and enhanced the reliability of the results.

Despite its strengths, this study has several limitations. The high heterogeneity observed across the included studies presents challenges to the generalisability of the pooled prevalence estimate. Variations in study design, diagnostic criteria, and population characteristics likely contributed to this heterogeneity. Additionally, the studies included in the analysis often had limitations in terms of sample size and representativeness, which could have influenced the accuracy of the findings.

This review exclusively considered studies published in English, potentially excluding relevant research published in regional or local journals that are not accessible through major academic databases. This restriction could have introduced sampling bias and limited the applicability of the findings to the entire Indian population. Many of the included studies focused on selected or specialised populations, further narrowing the scope of generalisation.

Publication bias among the selected studies was another concern as this may have affected the validity of the pooled CVD prevalence estimates. The methodological inconsistencies in defining CVD conditions across studies, along with the lack of standardised diagnostic criteria, add complexity to the interpretation of the results. Variance in the age groups studied and differences in study settings further reduce comparability and limit the conclusions that can be drawn.

These limitations highlight the need for future research to address gaps in knowledge and enhance the reliability of CVD prevalence estimates. Longitudinal studies with standardised diagnostic criteria, uniform definitions of CVD, and representative sampling are crucial for improving the accuracy and generalisability of prevalence findings in India.

## 6. Conclusions

This systematic review and meta-analysis provide valuable insights into the prevalence of cardiovascular diseases (CVDs) among adults in India, revealing a significant national burden of 11%. The findings highlight the rising prevalence of CVD, especially in urban areas, and reveal important regional and gender-based variations. Despite some heterogeneity across the included studies, the evidence underscores the escalating threat of CVD and emphasises the urgent need for public health interventions that address the impacts of urbanisation, lifestyle changes, and an ageing population. Additionally, raising public awareness about the widespread occurrence of cardiovascular diseases (CVDs) and their related risk factors is crucial. Education programmes focused on promoting healthier diets and lifestyles can play a key role in informing the public. A better understanding of these factors can aid in the early detection of CVD and contribute to the development of effective prevention strategies to address this growing health challenge.

## Figures and Tables

**Figure 1 ijerph-22-00539-f001:**
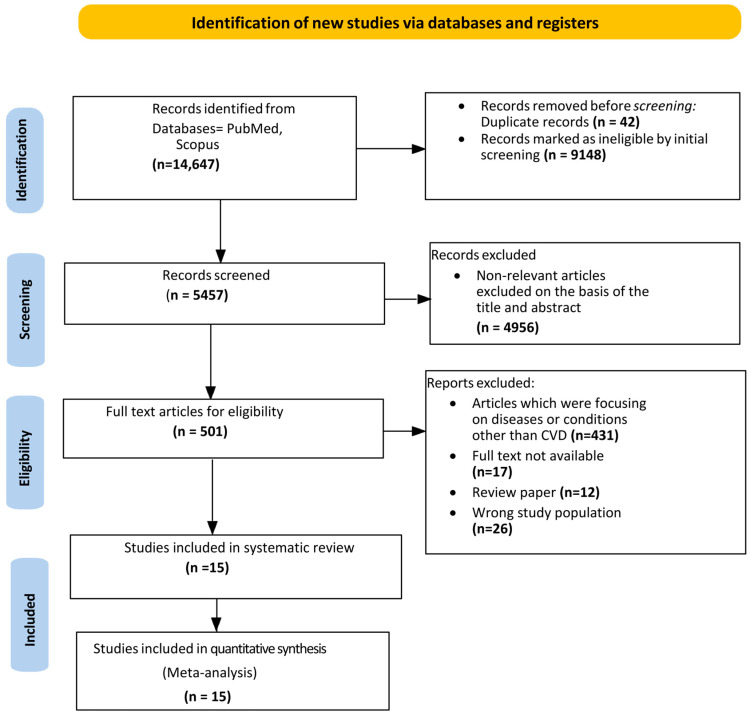
PRISMA flow chart of study selection.

**Figure 2 ijerph-22-00539-f002:**
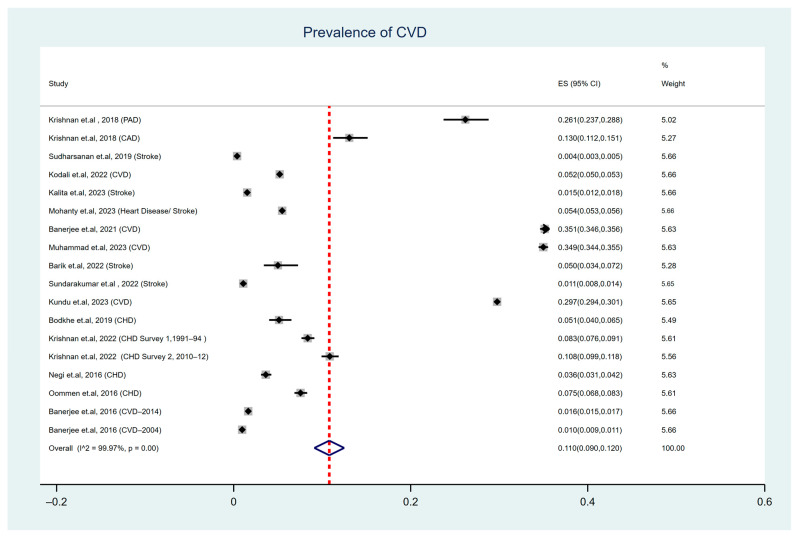
Forest plot illustrating the prevalence of CVD among the adult population of India [19,20,21,22,23,24,25,26,27,28,29,30,31,32,33].

**Figure 3 ijerph-22-00539-f003:**
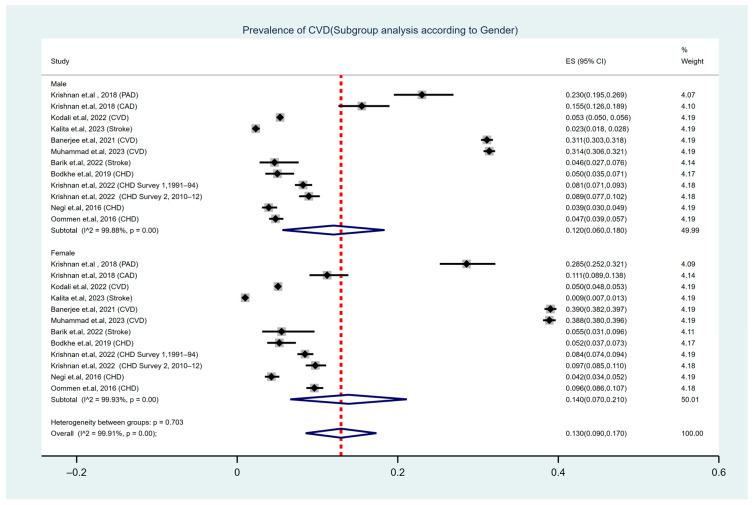
Forest plot illustrating the prevalence of CVD among the adult population of India, based on gender [19,21,22,24,25,26,29,30,31,32].

**Figure 4 ijerph-22-00539-f004:**
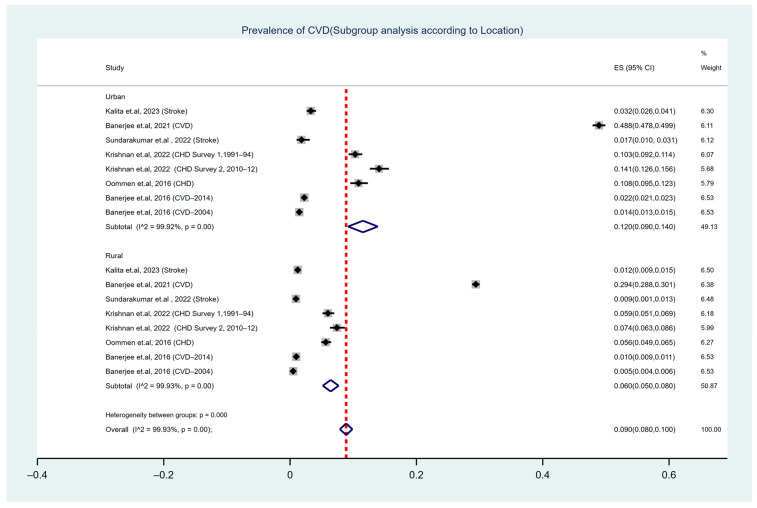
Forest plot illustrating the prevalence of CVD among the adult population of India, based on geographical location [22,24,27,30,32,33].

**Figure 5 ijerph-22-00539-f005:**
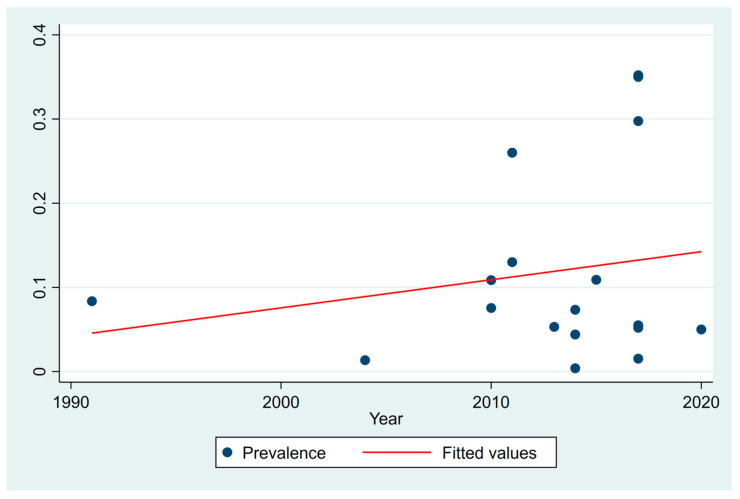
Prevalence of CVD over time in India.

**Figure 6 ijerph-22-00539-f006:**
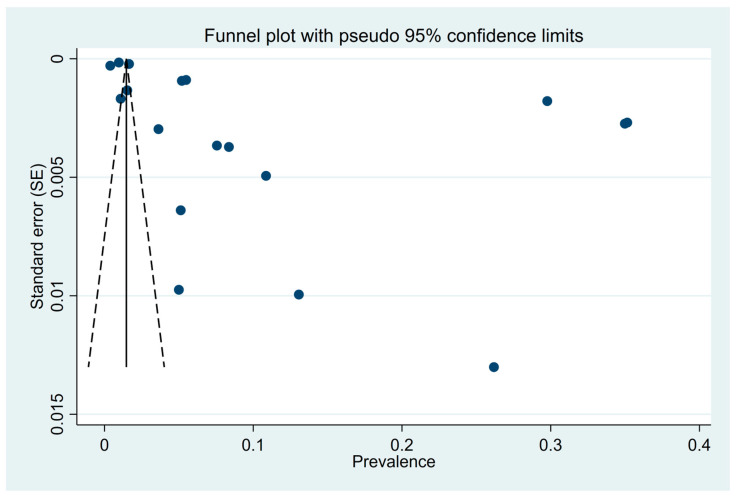
Funnel plot evaluating the prevalence of CVD in the Indian adult population for publication bias.

**Table 1 ijerph-22-00539-t001:** Characteristics of the studies on cardiovascular diseases (CVDs) in the Indian adult population.

Study	Year	Age Range	Gender	Sample Size	Study Area	Sampling Method	Study Design	Main Outcome	Criteria for Diagnosis of CVD	Prevalence (%)
Krishnan et al. [19]	2018	60–79 years	Both	1148	Northern parts of Kerela	Door to door	Cross-sectional	PAD and CAD	PAD by ABI, CAD by clinical history, and ECG	PAD = 26.7%, male = 23.01%, female = 28.57%; CAD = 13.07%, male = 15.52%, female = 11.20%
Sudharsanan et al. [20]	2019	60.9 years (mean age)	Both	45,053	Gadchiroli, Maharashtra	House to house	Cross-sectional descriptive study	Stroke	Confirmed by trained physician and neurologist using WHO’s criteria	Stroke = 388.4 per 100,000 person (0.39%), male = 519.4 per 100,000 person (0.51%), female = 255.1 per 100,000 person (0.25%)
Kodali et al. [21]	2022	≥45 years	Both	56,935	All Indian states and union territories	Multistage cluster sample survey	Cross-sectional	CVD, CHD, heart attack, and stroke	Clinical profile and medical history	CVD = 5.2%, female = 4.6%, male- = 5.9%; CHD/heart attack = 2.8%, female = 2.4%, male = 3.2%; stroke = 1.6%, female = 1.2%, male = 2.0%
Kalita et al. [22]	2023	≥45 years	Both	8496	Seven north-eastern states of India	Multistage stratified cluster sampling	Cross-sectional	Stroke	Clinical profile and medical history	Stroke = 1.53%, male = 2.3%, female = 1%, urban = 3.26%, rural = 1.21%
Mohanty et al. [23]	2023	≥45 years	Both	64,755	All Indian states and union territories	Stratified, multistage probability cluster random sampling	Cross-sectional	Heart disease/stroke	Clinical profile and medical history	Stroke = 5.5%
Banerjee et al. [24]	2021	≥60 years	Both	31,464	All Indian states and union territories	Multistage stratified area probability cluster sampling design	Cross-sectional	CVD	Clinical profile and medical history	CVD = 35.2%, female = 38.8%, male = 31.1%, urban—48.9%, rural—29.4%
Muhammad et al. [25]	2023	≥60 years	Both	30,333	All Indian states and union territories	Multistage stratified cluster sampling	Cross-sectional	Stroke, heart disease	Clinical profile and medical history	Stroke = 2.78%, male = 3.29%, female = 2.23%; heart disease = 5.25%, male = 5.81%, female = 4.64%
Barik et al. [26]	2022	60–100 years	Both	500	Bhubaneswar Odisha	Convenience sampling	Cross-sectional exploratory study	CVD	Clinical profile and medical history	CVD = 5%, male = 4.6%, female = 5.52%
Sundarakumar et al. [27]	2022	≥45 years	Both	3777	Karnataka	Multistage stratified	Cross-sectional	Stroke	Self-reported	Stroke = 1.09%, rural = 0.93%, urban = 1.80%
Kundu et al. [28]	2023	≥45 years	Both	65,562	All Indian states and union territories	Multistage stratified area probability cluster sampling design	Cross-sectional observational study	CVD	Clinical profile and medical history	CVD = 29.76%, poor = 24.6%, rich = 33.2%
Bodkhe et al. [29]	2019	≥60 years	Both	1190	Wardha district, Maharashtra	Not reported	Cross-sectional	CHD	Confirmed by health professionals	CHD = 5.31%, male = 5%, female = 5.25%
Krishnan et al. [30]	2020	35–64 years	Both	5535	Delhi NCR	Multistage cluster sampling (urban), random sampling (rural)	Cross-sectional	CHD	Minnesota-coded ECG	CHD = 8.36%, urban = 10.3%, rural = 6.0%, male = 8.21%, female = 8.44%
Krishnan et al. [30]	2020	35–64 years	Both	3969	Delhi NCR	Multistage cluster sampling (urban), random sampling (rural)	Cross-sectional	CHD	Minnesota-coded ECG	CHD = 10.86%, urban = 14.1%, rural = 7.4%, male = 8.9%, female = 10.2%
Negi et al. [31]	2016	20–70 years	Both	3968	Kinnaur, Himachal Pradesh	Simple random	Cross-sectional	CVD	By health supervisors	CVD = 4.4%, male = 3.9%, female = 4.5%
Oommen et al. [32]	2016	30–40 years	Both	6196	Rural and urban Vellore	Probability proportional to size (PPS)	Repeat cross-sectional	CHD	Previous diagnosis, ECG	CHD = 7.55%, male = 4.78%, female = 9.6%, rural = 5.66%, urban = 5.60%
Banerjee et al. [33]	2016	≥30 years	Both	385,055 (2004)	All Indian states and union territories	Systematic random sampling	Cross-sectional	CVD	Clinical profile and medical history	CVD = 0.734%, urban = 14.3%, rural = 4.86%
Banerjee et al. [33]	2016	≥30 years	Both	335,499 (2014)	All Indian states and union territories	Systematic random sampling	Cross-sectional	CVD	Clinical profile and medical history	CVD = 1.348%, urban = 22.44%, rural = 9.65%

Abbreviations: PAD, peripheral arterial disease; CAD, coronary artery disease; ABI, ankle–brachial systolic blood pressure index; ECG, electrocardiography; CVD, cardiovascular disease; CHD, coronary heart disease.

**Table 2 ijerph-22-00539-t002:** Risk of bias assessment following the Joanna Briggs Institute (JBI) Critical Appraisal checklist for studies reporting prevalence data [19,20,21,22,23,24,25,26,27,28,29,30,31,32,33].

Study	Was the Sample Representative of the Target Population?	Were Study Participants Recruited in an Appropriate Way?	Was the Sample Size Adequate?	Were the Study Subjects and Setting Described in Detail?	Was the Data Analysis Conducted with Sufficient Coverage of the Identified Sample?	Were Objective, Standard Criteria Used for Measurement of the Condition?	Was the Condition Measured Reliably?	Was There Appropriate Statistical Analysis?	Are all the Important Confounding Factors/Subgroups/Differences Identified and Accounted for?	Were Subpopulations Identified Using Objective Criteria?	Risk of Bias
Krishnan et al. (2018) [19]	Yes	Yes	Yes	Yes	Yes	Yes	Yes	Yes	Not clear	Yes	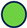
Sudharsanan et al. (2019) [20]	Yes	Yes	Yes	Yes	Yes	Yes	Yes	Yes	Not clear	Yes	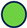
Kodali et al. (2022) [21]	Yes	Yes	Yes	Yes	Yes	Yes	Yes	Yes	Yes	Yes	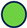
Kalita et al. (2023) [22]	Yes	Yes	Yes	Yes	Yes	Yes	Yes	Yes	Yes	Yes	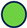
Mohanty et al. (2023) [23]	Yes	Yes	Yes	Yes	Yes	Yes	Yes	Yes	Yes	Yes	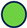
Banerjee et al. (2021) [24]	Yes	Yes	Yes	Yes	Yes	Yes	Yes	Yes	Yes	Yes	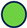
Muhammad et al. (2023) [25]	Yes	Yes	Yes	Yes	Yes	Yes	Yes	Yes	Yes	Yes	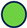
Barik et al. (2022) [26]	Yes	Yes	Yes	Yes	Yes	No	No	Yes	No	No	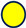
Sundarakumar et al. (2022) [27]	Yes	Yes	Yes	Yes	Yes	Yes	Yes	Unclear	Unclear	Yes	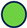
Kundu et al. (2023) [28]	Yes	Yes	Yes	Yes	Yes	Yes	Yes	Yes	Yes	Yes	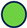
Bodkhe et al. (2019) [29]	Yes	Yes	Yes	Yes	Yes	Yes	Yes	Unclear	Unclear	Yes	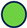
Krishnan et al. (2022) [30]	Yes	Yes	Yes	Yes	Yes	Yes	Yes	Yes	Yes	Yes	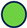
Negi et al. (2016) [31]	Yes	Yes	Yes	Yes	Yes	Not clear	Yes	Yes	Yes	Yes	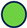
Oommen et al. (2016) [32]	Yes	Yes	Yes	Yes	Unclear	Yes	Unclear	Unclear	Unclear	Yes	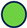
Banerjee et al. (2016) [33]	Yes	Yes	Yes	Yes	Yes	Not clear	Yes	Yes	Yes	Yes	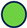

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
