# Peer review of "Prevalence of CVD Among Indian Adult Population: Systematic Review and Meta-Analysis"

_ijerph, 2025, doi:10.3390/ijerph22040539_

Round 1

Reviewer 1 Report

Comments and Suggestions for Authors

The study is quite interesting because it gives an overview of recent trends regarding CVD in India

Given that diabetes and hypertension are risk factors for cardiovascular diseases, is there a reason why papers discussing these conditions were excluded from the study?

Please highlight the significant lifestyle differences between India rural and urban areas in the introduction/background. Does the lifestyle difference or urbanization play a role in CVD prevalence?

Lines 250 and 251, Could the higher prevalence of CVDs in females be attributed to higher life expectancy among females? any thoughts or findings from other papers?

Comments on the Quality of English Language

The quality of English Language of this paper was good although there were some grammatical errors which I believe could be corrected with common word editing apps.

Reviewer 2 Report

Comments and Suggestions for Authors

I reviewed with interest the manuscript by Mohd. Shannawaz et al. "Prevalence of CVD among Indian Adult Population: Systematic Review and Meta-analysis". In this article, the authors attempted to estimate the prevalence of cardiovascular diseases in Indian residents by conducting a meta-analysis of observational studies. Some of these studies are based on large patient samples (up to 335,499  and 385,055), which is a strength of the analysis.

However, while reviewing the manuscript, I had some comments and questions, some of which are fundamental.

  1. The authors included studies with significant differences in their design in the meta-analysis. While some studies examined the prevalence of only individual nosologies (for example, Stroke or coronary heart disease according to the Minnesota ECG code), others included all cardiovascular diseases in the analysis. According to the authors, these were "coronary heart disease (CHD), cerebrovascular disease, peripheral arterial disease, rheumatic heart disease, congenital heart disease, deep vein thrombosis, pulmonary embolism" (lines 78-80). It is clear that the prevalence of individual nosologies was lower than the prevalence of ALL cardiovascular diseases. It is completely unclear how the average prevalence of diseases calculated on the basis of such studies can characterize the situation with morbidity in India. I am sure that if the studies that studied ONLY strokes or ONLY coronary heart disease in their cohorts assessed the entire spectrum of cardiovascular diseases, their prevalence would be closer to the figures obtained in the studies on those cohorts.
  2. The authors state in their article that "India has experienced the greatest increase in the prevalence of cardiovascular disease compared to any other country in the world [14,15]" (lines 58-59). In doing so, the authors refer to publications from 20 years ago. Perhaps this information was correct 20 years ago, but now the situation may have changed. More recent publications on this issue are needed.
  3. The authors write in the Introduction: "Detailed information on the prevalence of cardiovascular diseases (CVD) is crucial for planning and implementing effective preventive strategies. To address this knowledge gap, this systematic review and meta-analysis was conducted, using data from the scientific literature to provide a comprehensive insight into the prevalence of CVDs among the adult population of India. " (lines 63-67). But as I pointed out earlier, the design of the meta-analysis will not provide correct information on the prevalence of cardiovascular diseases. That is, the goal of the meta-analysis was ultimately not achieved.
Comments on the Quality of English Language

No comments

Reviewer 3 Report

Comments and Suggestions for Authors

Authors of this systematic review/meta-analysis report the prevalence of CVD in the Indian population. Understanding the prevalence of CVD is crucial to guide public health policies and research, thus of great importance to potential readers. However, several issues need to be addressed.

Major comments

  1. The inclusion criteria of CVD is very narrow. Although the WHO fact sheet mentions that 7 disease criteria is included in CVD, it doesn't necessarily mean that there's only 7 criteria in CVD. Other major CVD disease categories are aortic disease, arrhythmias, cardiomyopathies, non-rheumatic valvular disease, endocarditis, myocarditis, etc. The narrow inclusion criteria may underestimate the actual prevalence of CVD. Provide a reasoning for limiting the inclusion criteria or broaden it.
  2. The pooling method of overall CVD prevalence can be controversial. For instance, pooling PAD prevalence from study A with CVD prevalence from study B is problematic because PAD prevalence does not represent CVD prevalence. CVD prevalences (not with individual CVD) should be pooled or prevalence of each disease category should be each pooled.
  3. When pooling multiple studies, different patient characteristics should be taken into account. There is a significant variability in age range between included studies. It is clear from figure 1 that studies which included older patients have higher prevalence of CVD whereas studies which used a broader age range have lower prevalence of CVD. If the authors want to pool these studies to get an overall CVD prevalence, they should match the age criteria for each study since CVD prevalence increases with age.

Minor comments

  1. While it may be sufficient to access articles from pubmed and scopus, there are other databases such as EMBASE, web of science, medrxiv, Cochrane library, sciencedirect. It would be beneficial to search other databases as well for relevant articles.
  2. For figure 5, I suppose that the x-axis would be publication year. Use the actual study year, not the publication year. For example, 4 studies published in 2023 collected 2017-18 data. Thus, to identify if there's any time dependent increase in CVD rates, using study year, not publication year would be better.

Round 2

Reviewer 2 Report

Comments and Suggestions for Authors

The authors responded to my comments and made corrections to the text of the manuscript. I do not agree with the concept of this meta-analysis and the authors' efforts to overcome the heterogeneity of the population cohorts examined. However, if the editor deems it possible, the article can be published.